# Chlorophylls: A Personal Snapshot

**DOI:** 10.3390/molecules27031093

**Published:** 2022-02-07

**Authors:** Hugo Scheer

**Affiliations:** Bereich Systematik, Biodiversität und Evolution der Pflanzen, Universität München, Menzinger Str. 67, 80638 München, Germany; hugo.scheer@lmu.de

**Keywords:** chlorophylls, current research, applications, history

## Abstract

Chlorophylls provide the basis for photosynthesis and thereby most life on Earth. Besides their involvement in primary charge separation in the reaction center, they serve as light-harvesting and light-sensing pigments, they also have additional functions, e.g., in inter-system electron transfer. Chlorophylls also have a wealth of applications in basic science, medicine, as colorants and, possibly, in optoelectronics. Considering that there has been more than 200 years of chlorophyll research, one would think that all has been said on these pigments. However, the opposite is true: ongoing research evidenced in this Special Issue brings together current work on chlorophylls and on their carotenoid counterparts. These introductory notes give a very brief and in part personal account of the history of chlorophyll research and applications, before concluding with a snapshot of this year’s publications.

Chlorophyll (Chl) first came to me in a bottle containing approximately 1 kg of “Phäophytin Sandoz”, handed over by Hans-Herloff Inhoffen [1], the big boss of organic chemistry at the Technical University of Braunschweig. Herbert Wolf, one of the sous-chefs, had proposed using the Chl-macrocycle as a platform for studying the stereochemistry of side-chains for a diploma and later doctoral degree. The “Phäophytin Sandoz”, a mixture of (mainly) pheophytins *a* and *b* dating back from the time of Arthur Stoll [2] with Sandoz, proved an invaluable supply for making chemical modifications of the basic skeleton of chlorophyll *a*: after the first bottle ran out, it was replaced by another one and subsequently by a third one. Additionally, the spectroscopic tools available in Braunschweig, particularly circular dichroism and nuclear magnetic resonance, proved invaluable in analyzing the structure, including the stereochemistry, of the dozens of derivatives “cooked” from this material. As a postdoctoral researcher, I went to Joe Katz at Argonne National Laboratory. His group was famous for using the stable isotopes, ^2^H and ^13^C, for studying chlorophyll interactions in the test tube and in photosynthetic organisms [3]. I was lucky to have Jim Norris as a challenging guide. When we discussed his ideas of a “special pair” of Chls as the primary donor in photosynthesis, I remembered a selective deuteration from the Braunschweig lab that was key to proving the proposal by using Jim’s advanced electron double resonance tools [4].

This time as an apprentice solidified my passion in Chls and subsequently to open-chain tetrapyrroles. It provided the basis for studying, over the next 45 years, several aspects of this group of molecules under many guises. They provide the basis for photosynthesis and thereby most life on Earth, in addition to several other functions. Besides their involvement in primary charge separation in the reaction center, they serve as light-harvesting and light-sensing pigments, they also have additional functions, e.g., in inter-system electron transfer. Chls also have a wealth of applications [5,6]: in basic science, modified pigments allow for functional analyses; in medicine, they serve as photosensitizers. Chl derivatives are used as colorants for food and cosmetics, they are also candidates for optoelectronic use. Much of the work performed in Braunschweig, Argonne, and later on in Munich refers to photosynthesis, although this was rather detached from living organisms. The extent of how detached it became is illustrated by two incidents. The first was when I handed my doctoral thesis to my parents. They were quite startled to see a such a tiny booklet of approximately 150 DIN A5 pages as the result of three years of work. Moreover, they confessed that they hardly understood anything besides the introductory sentences. The second incident was when my young children accompanied me one day to the basement at Argonne where all kinds of organisms were grown on stable isotope media. When I was shaving a plate of two-week-old wheat, their eyes lit up: for the first time, there was something familiar to them in the labs. Since then, I was a farmer of baby corn to them. It soon became obvious to them, however, that things really became interesting for me once the corn, the algae or the bacteria were crushed and their cells broken up.

It was satisfying to contribute during this time to the accumulated knowledge of tetrapyrroles, with the help of many students and technicians, in addition to funding from a variety of institutions of which the Deutsche Forschungsgemeinschaft was the most prominent one, in cooperation with groups from many countries of whom many become friends for life. The story of chlorophyll research goes back more than 200 years. The green extract of green plants and algae, then termed chlorophyll (Greek: green of leaves) by Pelletier and Caventou [7], later turned out to be a mixture of two non-fluorescent orange-yellow pigments and two blue tetrapyrrole pigments that fluoresce brightly [8]. While Stokes only gave incomplete details on how he came to this conclusion [9], it was verified by Fremy [10] and subsequently by Tswett [11], who coined the term chromatography for the novel separation developed with these pigments. The yellow pigments are carotenoids. The term chlorophyll has been narrowed to the family of tetrapyrrole pigments [12] which in plants are the blue Chl *a* and Chl *b.* The chemistry of these chlorophylls was studied early on by Willstätter’s group [13] who realized their relation to heme. They also studied chlorophyllase and its hydrolyzing action on Chls. While it was one of the first enzymes to be isolated, its function in vivo remains enigmatic today [14]. The molecular structure of Chls *a* and *b* was solved by the group of Hans Fischer in 1942 [15]. Their stereochemistry was determined by Ian Fleming, by Burrell et al. and by Crabbé et al. (reviewed in Brockmann, [16]). Total syntheses were achieved by the groups of Strell [17] and, more rigorously, Woodward [18], but a general synthetic access of Chl-type pigments remains a challenge [19,20].

Chl-type pigments means that there are more than just Chls *a* and *b*. Stokes already mentioned a third chlorophyll present in brown algae, again without giving details [8]. Today, this is recognized as a mixture of several *c*-type chlorophylls [21]. Three more chlorophylls (Chls *d*, *e*, *f*) have been described in organisms capable of oxygenic photosynthesis [22,23,24], of which only the structures of Chl *d* [24,25] and Chl *f* [26] have been determined. They are supplemented by small amounts of especially tailored derivatives: the so-called prime pigments, isomers at C-13^2^ of the parent compounds [27], the demetalated pheophytins, and pigments carrying alcohols different from the ubiquitous isoprenoid, phytol [27,28]. A much larger number of Chls, more precisely bacteriochlorophylls (BChls), has been found in photosynthetic bacteria [29]. Altogether, more than 100 (B)Chls are known in plants, algae and photosynthetic bacteria (Figure 1). A common feature of all is a long-lived excited singlet state, S_1_, that in a monodisperse solution, results in bright fluorescence, while intersystem crossing to the phototoxic T_1_ state is relatively low [30]. Their common structure is a cyclic tetrapyrrole with an additional isocyclic ring and, with only a single rare exception [31], a central metal Mg^2+^-ion. While the c-type Chls are true porphyrins, all others are chlorins reduced at ring D or bacteriochlorins reduced at rings B and D. These reductions result in an increased absorption and a red-shift of the redmost absorption band (Q_Y_-band) that extends photosynthesis into the near-infrared spectral range down to 1050 nm.

Regarding the long history of chlorophyll research [15,22,32,33,34,35], one would think that all has been said on these pigments. However, the opposite is true. Just by scanning through the publications of recent years, SciFinder lists 43,314 references since 2016. Additionally, while writing this preface, there have been publications, on an almost daily basis, on the various aspects of (B)Chls that witness unabated interest in these fascinating pigments and keep me reading on the various aspects of their structures, functions and uses. Moreover, ongoing research evidenced in this Special Issue brings together current work on Chls and on their carotenoid counterparts.

Let me illustrate this, before closing these notes, with a snapshot of publications from this year that refer to the various aspects of Chl research and applications. There is, firstly, the molecule itself. For better understanding the aromaticity of the parent chlorin and bacteriochlorin structures, their π-systems were compared to porpholactones, oxa-derivatives of chlorins and bacteriochlorins carrying a peripheral C=O group next to the oxa-O [36]. Compared to the fully unsaturated porphyrins, the ring-current in Chls and BChls bypasses the reduced peripheral positions. However, the π-density is partially restored in these formal single-bonds by the participation of non-bonding electrons of both oxygens. Another structural feature of most Chls is the isoprenoid alcohol esterified to the C17-substituent. The spectroscopic influence of the alcohol is distinct but minor [37], however, the alcohol is generally conserved among the different groups of photosynthetic organisms. In green plants, it is generally phytol. Increasing amounts of the precursor, geranylgeraniol, are found in *Arabidopsis thaliana* after growth in green light [38]. The optical spectra of (B)Chls have been studied from the beginning. However, even these are open for surprises, as Leiger et al. [39] showed in their contribution to this Special Issue. They showed by anti-Stokes fluorescence that the red-most absorption band (Q_Y_) has, at ambient temperature, has a tail extending far into the near-infrared. They ascribe it to the thermally activated vibronic coupling of the electronic transition. While the extreme red-shifts extending photosynthetic activity to lower energies can be ascribed in several cyanobacteria to chemically modified Chls *d* and *f* (see below), many organisms containing only Chl *a* show similar or even larger red-shifts; a prominent example is *Ostreobium* [40,41]. It is conceivable that the intensity of the band studied by Leiger et al. [39] is increased in these organisms by interactions with the protein. Chl cation radicals are formed in the primary reaction of photosynthesis by electron transfer. Orzel et al. [42] have now shown that in the presence of CuCl_2_, they can also be generated and participate in electron transfer in the dark.

Correlating function and spectroscopy to individual chlorophylls in native protein structures such as photosystem I remains a challenge. By polarized single-molecule spectroscopy, the individual Chls of the 96 present in the structure have been related to pigments absorbing at the blue and red edge of the spectrum [43]. In particular, the latter are relevant in localizing excitation energy near the reaction center. In this case, the red-shift is due to the still poorly understood interactions with the environment. In other cases, it relates to chemically modified pigments compared to the ubiquitous Chl *a.* Chls *d* and *f* carry peripheral formyl- instead of methyl-substituents at rings C and A, respectively (Figure 1) [24,44]. Such red-shifted Chls are relevant for cyanobacteria extending the hitherto-thought energetic limits of oxygenic photosynthesis [45]. In one such species, *Synechococcus thermalis* PCC 7335, their locations in photosystem I were identified by cryo electron microscopy: one Chl *d* is part of the primary donor and two Chl *f* are part of the core antenna [46]. A huge Bchl-complex largely devoid of protein is the chlorosome of green bacteria. Its structure is largely determined by BChl–BChl interactions between members of the BChl *c*, *d* and *e* family [47,48,49]. The group of Jürgen Köhler has used linear dichroism of single light-harvesting complexes from purple photosynthetic bacteria to obtain information on the pigment arrangement and its excitonic coupling [50]. In their contribution to this issue, Günther et al. [51] concluded that the powerful method reaches a limit with complexes of the size of the chlorosome of *Chlorobaculum tepidum*

The long lifetimes of excited singlet states (S_1_) of Chls are crucial to their function in photosynthesis. They also pose problems to photosynthetic organisms, however, because they increase the chance for intersystem crossing to the triplet state (T_1_) that subsequently generate singlet oxygen. Even though this process is minimized in the Mg-complexes [30], Chls are highly phototoxic under conditions of over-excitation of the photosynthetic apparatus when the productive use of the excitation energy is limited. A large fraction of the photosynthetic apparatus is therefore devoted to photoprotection [52], with carotenoids serving several lines of defense. The publication of Demming-Adams et al. in this Special Issue is devoted to a particularly effective carotenoid, zeaxanthin [42]. The role of carotenoids is, however, multi-faceted. As reviewed by Lokstein et al. in this issue [49], they often participate in light-harvesting, transferring their excitation with good to excellent quantum yields to Chls. Additionally, Makhneva et al. [53] presented an example from purple bacteria where the protective function against one ROS, e.g., singlet oxygen, seems negligible.

Phototoxicity also becomes relevant if (B)Chls are detached from their native protein complexes or during biosynthesis and biodegradation. The protection of phototrophic organisms during the metabolism of chlorophylls is ascertained by a variety of mechanisms. During biosynthesis, several control mechanisms have been recognized [54]: (i) the complete macrocyclic porphyrin system is only generated in the very last and tightly controlled step, all precursors are non-fluorescent and thereby non-phototoxic; (ii) the very late precursor is incorporated, in oxygenic organisms, in a light-activated reductase; (iii) once formed, Chls are bound to proteins which also contain carotenoids as protective pigments. During biodegradation, the macrocycle is broken down after demetalation and dephytylation to non-phototoxic open-chain tetrapyrroles, the phyllobilins, on which all subsequent modification and degradation steps are performed [55]. Phototoxicity is also a potential problem in photosymbiotic partnerships, particularly for organisms harboring stolen and still active chloroplasts for several months (kleptoplasty) [56,57]. Last but not least, the phototoxicity of (B)Chls is a problem for herbivores including humans. Higher animals degrade Chl and its derivatives in their intestine [58] and have a low intestinal uptake [59]. This is combined with (at least) one excretory system; Szafraniec and Fiedor et al. [60] have now studied its specificity towards a variety of chlorophyll derivatives. Lower animals have developed a different strategy: they modify the structures by cyclization between the C-18 propionic side chain and C-13^2^ of the isocyclic ring, thereby generating C-13^1^ enols that are no longer phototoxic [61]. A massive reduction in excited-state lifetime has been known for peripheral metal complexes containing this structural element [62] and recently demonstrated as a characteristic feature of cyclo-enols [63].

The most widespread application of Chls are assays in plant biology. Often, the pigments are extracted and then spectrophotometrically analyzed [64,65]. Alternatively, the absorption or emission of (B)Chls is used for monitoring on time scales ranging from nanoseconds to years and spatial scales ranging from single molecules to the whole Earth or even outer space. Several reviews published in 2021 summarized current developments [66,67,68,69,70,71,72]. While these applications used the pigments as biological reporters, a current publication investigates the use of Chl *a* as an x-Ray dosimeter [73].

Moreover, there are much wider applications, probably the most advanced being their use as photosensitizers in medicine [74,75,76,77]. Their phototoxicity has generated interest in their use to tackle unwanted cells, from harmful microorganisms to tumors. Suvorov et al. reviewed the photodynamic therapy directed against microbes [74]. The capacity of porphyrins for generating singlet-oxygen has been used for almost 50 years in tumor therapy [75]. Phototoxicity is enhanced in chlorins and all the more in bacteriochlorins [78,79,80] due to increased absorption and a shift to longer wavelengths penetrating deeper into tissue. Moreover, when using Pd-derivatives of BChl, it became clear that additional reactive oxygen species are involved and indirect modes of attack are relevant. They are currently implemented in the treatment of pre-clinical prostate tumors [79]. Other medical applications are being developed in ophthalmology, a current example being cornea cross-linking [80]. Particularly elegant is it their use in theranostics, in which phototoxic therapy is combined with diagnostic mapping, using either the same molecule for both purposes, or conjugates. A current publication on the latter uses a combination of a photosensitizing bacteriochlorin with naphthylimide fluorophores [81]. A general problem of using Chls in many such applications is their low stability and poor solubility. One of the solutions to overcome this is by solubilizing the pigments in detergent micelles. In their contribution to this issue, Janik-Zabrotowicz et al. [82] described the stabilization of Chl over extended times against light and oxygen by solubilizing it in polyoxyl 35 Castor oil (Chremophore EL) that forms exceptionally small micelles. It contains the pigment in monomeric form, thereby avoiding unwanted effects due to aggregation. Another common way is the introduction of polar side-chains that is facilitated by the functional peripheral substituents of (B)Chls. Again, only a current example shall be mentioned that uses polyethylene-glycol for this purpose [83].

The bright color of Chls in a range not covered by other natural dyes has led to many applications as colorants [6]. The additive E140 is regarded as chlorophyll, although it contains mainly oxidation products that retain the central Mg^2+^ ion and the lipophylic esterifying phytol (personal observation, unpublished). E140, E140i and E140ii have the central Mg^2+^ replaced by Cu^2+^, thereby reducing its phototoxicity. In recent work, Perez-galvez et al. [58] studied the composition of E140i, i.e., a lipophilic [Cu]–Chl preparation, and its degradation in the intestine. Last but not least, there is also work for stabilizing Chl (or at least derivatives retaining its color) for use in paints [84]. Vegetables and green algae are a source of a mixture of Chl *a* and Chl *b.* Cyanobacteria are a versatile source for pure Chl a and its derivatives. Depending on the growth conditions [85], they also yield the blue, water-soluble phycocyanin carrying open-chain tetrapyyrole chromophores. In my group, we kept batches rich in Chl *a* and batches rich in phycocyanin as ready and easily stored supplies for these pigments for subsequent derivatization.

Much less developed are applications in the very realm of Chls, i.e., photovoltaics. While simple devices coined, e.g., synthetic leaf based on Chls are more than 50 years old, these pigments have thus far turned out too short-lived for applications, with the possible exception of chlorosome-like Chl aggregates [86]. The largest impact from natural photosynthesis is conceptual: the combination of reaction centers for charge separation with light-harvesting complexes that are adaptable to the quality and quantity of absorbed light is key to natural photosynthesis. Prototypes of photovoltaic devices applying this concept via dye-sensitized cells and, possibly, nano-particles relying on absorption enhancement by nanoparticles, reference some of the current reviews which conclude this personal snapshot [87,88,89].

## Figures and Tables

**Figure 1 molecules-27-01093-f001:**
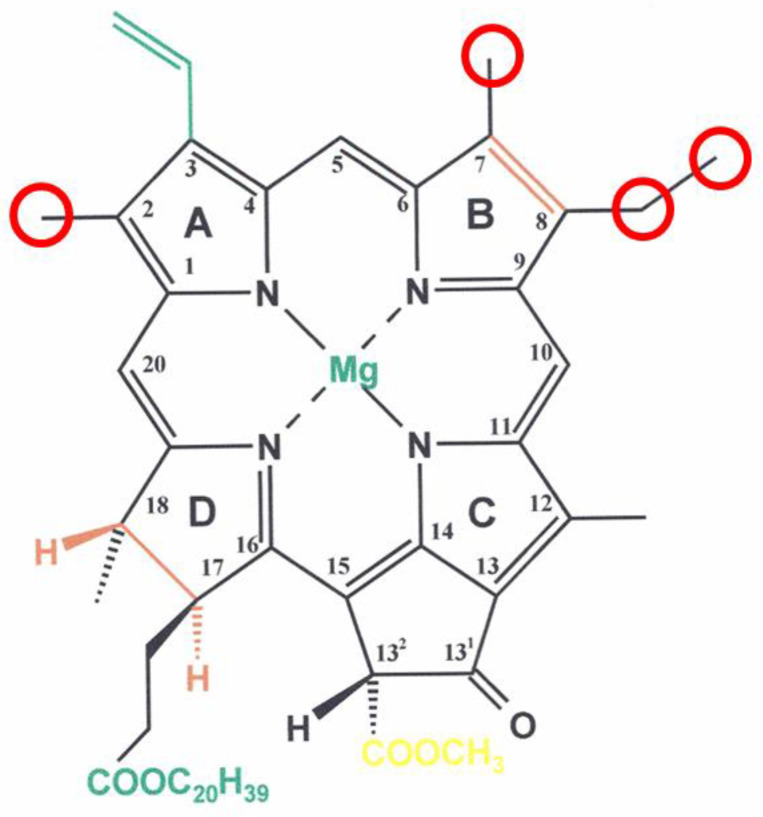
Structure of Chl *a.* Circles and colored bonds indicates modified sites in other Chls and BChls. Chl *b* and Chl *f* have a 7- and 1-CHO group, respectively_._ BChls *c, d* and *e* are characterized by lacking the 13^2^-COOCH_3_, by a CHOH-CH_3_–group at C2 (both stereoisomers), they differ by methylations at C20 and at the C8 and C12 substituents. In BChls *a, b* and *g,* the 7,8-double bond replaces the single bond, they differ in the substituents at C3,7 and/or 8. The c-type Chls are characterized by a 17,18-double bond, they differ by further modifications. Additional modifications in certain (B)Chls are replacements of the phytol esterifying the 17–propionic acid by other alcohols, mainly derived from farnesol or geranylgeraniol, exchange of the central Mg^2+^ by Zn^2+^ or inversion of the stereochemistry at C13^2^. For further readings, see Ref. [24].

## Data Availability

This work does not contain original data.

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
