# Peer review of "Chlorophylls: A Personal Snapshot"

_molecules, 2022, doi:10.3390/molecules27031093_

Round 1

Reviewer 1 Report

Comments and Suggestions for Authors

The work is very interesting and offers data of great interest.

Abstract

Line 10 and 11

After the abstract

Please provide abbreviations

Please follow the comments in the pdf version as follow:

  • Lines 42-48
  • Lines 63, 64, 72, 73
  • Lines 106, 107, 108, 115, 125
  • Lines 154, 173, 186, 187
  • Lines 203, 213, 218
  • Lines 221 and all references

References

Please follow the Journal roles in references

Best regards

Author Response

Reviewer no. 1

Thanks for the careful reading and valuable suggestions. Most of them I have taken as suggested, with the following exceptions (all numbers refer to the lines in the updated manuscript):

41 ff This sentence has been rephrased and corrected. The optoelectronic potential has been removed, it is still wishful thinking as far as I am aware.

47 ff This sentence has be rephrased, too.

67 This reference is in German. I prefer to keep it nonetheless because of its relevance

74 I have added Brockmann as reviewer of the stereochemical work and author of ref 14.

76 May I keep both, the names of Strell and of Woodward. There has been a controversy as to whether Strell’s work is considered a total synthesis, but agreement about the achievement of Woodard’s group.

References: Refs 44 and 45 need to be combined into one. Due to the resulting „frame shift“, the following references had been misnumbered; this has been mended. Additional changes in the numbering result from newly introduced references as suggested by other reviewers. As a result, most of the references are renumbered.

Reviewer 2 Report

This personal snapshot could be an interesting historic contribution to the chlorophyll research. Although the reference to many applications of this molecule it seems to me that some very current approaches in terms of photosynthetic symbiosis. namely functional kleptoplasty in Metazoa, must be referred  (please see, as an exemple, doi:  10.1098/rspb.2021.1779)

Unreliable bibliographic references, such as those from Wikipedia, should be avoided.

Author Response

Thanks for pointing out kleptoplasty in Metazoa. It certainly adds to the problems enacountered by non-photosynthetic organisms when coping with chlorophylls. Refs. 52X and 53x have been added to the topic.

You are right in avoiding references like Wikipedia. Refs. 1 and 2 have been replaced, accordingly, by more stable sources.

Reviewer 3 Report

This "perspective" on chlorophylls is an excellent text, very informative, easy to read, providing abundant details about these pigments, some of which (especially the history) I confess I was unaware of.

It succinctly describes the author's extensive knowledge of chlorophylls and their high importance in the contexts of photoautotrophic organisms.

The text describes the historical perspective of the discovery and characterization of the different chlorophylls, the chemical structure, the link between structure and function, the issues not yet fully discovered, phototoxicity and the applications of these molecules.

I believe that the text is of high quality and therefore deserves to be published in Molecules.

I have only two remarks:

  1. The chlorophyll molecules could be illustrated, to make it easier to read the molecular differences between them;
  2. the author did not intend, I believe, to be very exhaustive on the biotechnological applications of chlorophylls, since many others remain to be mentioned (cosmetics, pigments/colorant for textile, food additive with code E140, animal feed, antioxidant, anti-obesity, to name just a few).This issue should be clarified.

Two small additional corrections:

Line 46 – "they" instead of “the”

Line 94 – “that” is duplicated

Author Response

Thank you for the review and for your suggestions which are taken into account in the revision.

1: I have added early on a single structure showing chlorophyll a, in which all structural elements that differ from it in other chlorophylls and bacteriochlorophylls have been highlighted. For a short overview, this seemed to me more instructive (and compact) than giving a dozen or more stuctures. I hope this is acceptable.

2: The use of chlorophyll derivatives as colorants has been added as a separate paragraph.

Round 2

Reviewer 1 Report

I recommend the publication in the present form